# Influence of Variety, Enzyme Addition and Destemming on Yield and Bioactive Compounds of Juices from Selected Hybrid Grape Varieties Cultivated in Poland

**DOI:** 10.3390/foods12183475

**Published:** 2023-09-19

**Authors:** Muhamad Alfiyan Zubaidi, Marta Czaplicka, Joanna Kolniak-Ostek, Agnieszka Nawirska-Olszańska

**Affiliations:** 1Department of Fruit, Vegetable and Plant Nutraceutical Technology, Wrocław University of Environmental and Life Sciences, 51-630 Wrocław, Poland; muhamad.zubaidi@upwr.edu.pl (M.A.Z.); joanna.kolniak-ostek@upwr.edu.pl (J.K.-O.); 2Department of Horticulture, Wroclaw University of Environmental and Life Sciences, 50-375 Wrocław, Poland; marta.czaplicka@upwr.edu.pl

**Keywords:** hybrid grape, grape juice, enzyme addition, destemming, bioactive compounds, polyphenols, antioxidant capacity

## Abstract

In this study, five varieties of hybrid grapes were processed into juice to assess the influence of variety, destemming process and enzyme addition on juice quality, including yield, physicochemical properties and bioactive compounds. The results highlighted that while the processing methods had some impact on juice quality, the inherent grape variety remained the most significant factor. Although similar treatments were applied to all varieties, there were differences in the quality attributes of the juice. In general, red grape juice had a higher polyphenol content and antioxidant capacity than white grape juice. Four phenolic acids, eleven flavonols, five flavan-3-ols and five anthocyanins were identified. While the polyphenolic profile of each sample depended on the treatment and the variety, regardless of the variety, destemming was found to increase the yield by around 10–15%, while the addition of enzyme improved the yield by around 20–30%. Although the addition of enzymes led to a higher polyphenol content, it changed the color profile of the juice as a result of the pigment’s skin extraction. In contrast, the destemming process did not influence the color of the juice, but at the same time, it reduced the health benefits due to the removal of potential bioactive compounds from the stems.

## 1. Introduction

Grapes are good sources of antioxidants and bioactive flavonoids and offer numerous health-promoting properties with a rich taste and aroma. The antioxidant activity of the polyphenols present in grapes is believed to be much higher than that of essential vitamins and contributes to the dietary benefits of grapes [1].

Grapes have a high polyphenol content, including flavonoids (anthocyanins, flavonols and flavan-3-ols), phenolic acids, triterpenes and minerals that benefit the human body. These bioactive compounds possess free radical scavenger properties that protect the circulatory, nervous and immune systems and maintain significant health functions [2,3].

Nowadays, viticulture is mainly associated with countries with warm climates and long, hot summers. However, vineyards have also settled in countries with less favorable and cold climates such as Poland. According to Council Regulation (EC) 2165/2005, Poland is included in Zone A, the coldest zone of the winegrowing region [4].

Climate is undoubtedly one of the most critical factors influencing grape cultivation quality and productivity. With respect to the viticulture theory, two types of climate are most often distinguished: warm and cold [5]. Cold climate refers to regions where the average month’s temperature preceding the grape harvest season is less than 15 °C. The temperature usually drops quickly in the cold climate region before harvest, making the grapes have different tastes and flavors. Low temperatures preserve acidity in grapes, which can result in grapes with higher acidity levels. However, excessive acidity may not be preferable to consumers. In a cold climate like Poland, viticulture is more challenging than in warm regions. Grapes are susceptible to winter frost and fungal disease. Due to weather conditions, the berries may not reach their complete maturation composition [6]. The conventional variety of grape (*Vitis vinifera *L.) is poorly adaptive to fungal infection and winter frost, making it unsuitable to grow in cold climates. Therefore, a hybrid grape, a cross-species variety, is introduced to improve its resistance to fungal disease and low temperatures [7,8]. However, hybrid grape varieties are characterized by thick skin and have a high polyphenol content [3,9]. This character makes the hybrid grape less preferable for direct consumption. Consequently, over 80% of grapes grown worldwide are used for wine or juice manufacturing [10].

Several previous studies on hybrid grapes grown in Poland investigated mainly wine [9,11,12,13], but no research was found investigating juice processing using hybrid grapes grown in Poland as raw materials. Therefore, the investigation of hybrid-grape juice processing will provide significant knowledge that can be contributed by this study.

In industrial grape juice processing, enzymes play a crucial role in enhancing both the yield and quality of the juice. The addition of enzymes catalyzes the degradation of cell walls, thereby facilitating juice extraction and resulting in a higher yield. Furthermore, the introduction of enzymes aids in the extraction of phenolic compounds, color pigments, aromas and other soluble constituents [14].

Souquet et al. reported that grape stems contain a significant amount of bioactive compounds [15]. The incorporation of grape stems into the maceration process is expected to produce juice with a higher polyphenol content. This study aims to investigate the impact of grape variety, enzyme addition and destemming processes on yield, physicochemical properties, total phenolic content, antioxidant capacity and individual phenolic compounds in grape juice.

## 2. Materials and Methods

### 2.1. Grapes

Three white grape varieties (Johanniter (J), Muscaris (M) and Aurora (A)) and two red grape varieties (Golubok (G) and Regent (R)) were obtained from the Research and Teaching Station of the Department of Horticulture of the Wroclaw University of Environmental and Life Sciences in the village of Samotwór, Wroclaw, Lower Silesia Province, Poland. The characteristics of each cultivar are represented in Table 1.

### 2.2. Reagents, Enzymes and Standards

The standards caffeic acid, caftaric acid, coutaric acid, fertaric acid, kaempferol-3-O-rutinoside, quercetin-3-rutinoside, kaempferol 3-O-galactoside, quercetin-3-glucuronide, quercetin-3-galactoside, quercetin-3-glucoside, quercetin-3-rhamnoside, kaempferol-3-O-glucoside, isorhamnetin 3-glucoside, dihydroquercetin-3,5-rhamnoside, dihydrokaempferol-3-glucoside, procyanidin trimer, procyanidin B2, procyanidin tetramer, (−)-epicatechin, (−)-epicatechin 3-gallate, petunidyn-3-O-glucoside, malvidyn-3-O-glucoside, delphinidyn-3-O-(6′-acetyl)-glucoside, petunidyn-3-O-(6″-p-coumaroyl)-glucoside and malvidyn-3-O-(6″-caffeoyl)-glucoside were purchased from Extrasynthese (Lyon Nord, France). 2,2′-Azino-bis (3-ethylbenzothiazoline-6-sulfonic acid, ABTS), gallic acid, FeCl_3_, Trolox and all solvents were purchased from Sigma-Aldrich (Steinheim, Germany). Pectinex^®^ Ultra SP-L was obtained from Novozymes (Bagsvaerd, Denmark).

### 2.3. Juice Processing

The grapes were kept in chilling conditions at 4 °C until juice processing. The grapes were processed into juice within a maximum of five days after harvesting. The grapes were washed and destemmed if necessary.

Before macerating at 40 °C for 90 min, the grapes were crushed and 0.05% (*w*/*w*) of Pectinex^®^ Ultra SP-L enzyme was added. The must was then pressed using a hydraulic press (type TPZ 7, Bucher-Guyer, Niederweningen, Switzerland) and filtered to obtain juice. The juices were immediately frozen at −20 °C until analysis. The samples according to the varieties and treatments are shown in Table 2.

### 2.4. Dry Matter, Total Soluble Solid, Titratable Acidity and Pectin

The determination of the dry matter content (dm) of the juices was carried out in a vacuum dryer (SPT-200, ZEAMiL Horyzont, Kraków, Poland) at 80 °C for 24 h at a pressure of 1 kPa. The measurement was carried out in duplicate.

The total soluble solid was measured using a Refractometer PAL-1 (Atago, WA, USA). The measurement was taken in duplicate and expressed as °Brix.

The titratable acidity (TA) was determined according to titration aliquots using a titrator (Orion Star T910, Thermo Fisher Scientific, Beijing, China) and following the Polish standard (PN-90/A-75101/04, 1999) [16]. NaOH (0.1 N) was slowly added to the whole fresh grapes and homogenized until the endpoint (pH 8.1), and this process was controlled using an automatic pH titration system. The measurement was carried out in triplicate and expressed as g of tartaric acid per 100 mL. The analysis of pectin content was performed according to Pijanowski et al. [17].

### 2.5. Color (L*a*b)

The color of the grape juices was measured with a Color Quest XE Hunter Lab colorimeter (Reston, VA, USA) using CIE standard Illuminant D65 at 10° observer. The measurement was performed in triplicate. The results of the color measurements were expressed as L*a*b values (L (lightness), a (red–green) and b (yellow–blue) [18].

### 2.6. Turbidity and Viscosity

Juice turbidity was measured with a turbidimeter (Turbiquant 3000 T; Merck; Dramstad, Germany). Turbidity was measured in duplicate and expressed as NTU. The viscosity was determined using a viscometer (DV-II+ PRO VISCOMETER; Brookfield, Harlow, UK) at ambient temperature. The viscosity measurement was performed in triplicate and expressed as mPa·s.

### 2.7. Antioxidant Capacity

The extraction of the sample for antioxidant capacity and polyphenol content was carried out according to Kolniak-Ostek [19]. The antioxidant capacity of the grape juice was determined by ABTS and FRAP using a Synergy H1 spectrophotometer (BioTek Instruments Inc., Winooski, VT, USA) as previously described by Re et al. [20] and Benzie and Strain [21], respectively. The measurements were carried out in quadruplicate, and the results were expressed as mmol of Trolox equivalent (TxE) per liter of juice.

### 2.8. Total Phenolic Content

The total polyphenolic content was determined in 70% methanol extract (*v*/*v*) using the Folin–Ciocalteu method and measured using a Synergy H1 spectrophotometer (BioTek Instruments Inc., Winooski, VT, USA). The measurement was carried out in quadruples, and the result was expressed as g of gallic acid equivalent (GAE) per 100 mL of juice.

### 2.9. Identification and Quantification of Polyphenols

The polyphenols in the samples were identified and quantified using UPLC-PDA-FL (Waters Corp., Milford, MA, USA). The preparation and analysis of the sample were carried out as described by Nawirska-Olszańska et al. [22]. A sample of 5 μL was taken by an autosampler (BEH C18 column, 2.1 × 100 mm, 1.7 µm; Waters Corp.; Dublin, Ireland) at 30 °C and gradient elution (0–12 min 98–65% A, 12.1–13.5 0% A, 13.6–15 min 98%) with the mobile phase of solvent A (0.1% formic acid in water) and solvent B (0.1% formic acid in acetonitrile). The separation parameter of the mass spectrometer followed Kolniak-Ostek and Oszmiański [23]. The identification of polyphenols was determined by the retention time (Rt) of the standards. The polyphenol content was quantified using calibration curves of selected pure compounds.

### 2.10. Statistical Analysis

Statistical analyses were carried out using Statistica 14 (Statsoft, Tulsa, OK, USA). The factorial analysis of variance (ANOVA) with the HSD Tukey post hoc test (*p* < 0.05), cluster analysis and correlation matrix were performed to compare the results.

## 3. Results

### 3.1. Yield, Initial Moisture Content and Pectin Content

A statistical analysis indicated that both processing technology and variety have a significant impact on the juice yield. Figure 1 illustrates the yield of each variety under different treatments, along with the initial moisture and pectin content.

The enzyme addition and destemming processes significantly enhanced the processing efficiency. In most cases, both treatments led to an increase in yield of around 20–30% and 10–15%, respectively. The enzymes effectively degraded pectin, facilitating the easy release of juice from the fruit, resulting in a higher yield [24]. As highlighted in a previous study conducted by Guerrini et al. [25], the destemming process reduced the required pressure for juice extraction, particularly when the grapes had already been crushed. Subsequently, it led to the extraction of a greater amount of juice under comparable working pressures.

Despite employing similar procedures, there was a notable disparity in yields among different varieties. This indicated that the characteristics of the raw material had a key influence on the juice yield. The moisture and pectin content of the raw materials were significant factors influencing the juice yield. A correlation analysis revealed a positive correlation between yield and moisture content, while a negative correlation existed between yield and pectin content. Pectin is crucial to determining the firmness of the fruit and the ease of juice release. Consequently, fruits with a higher pectin content may experience a reduction in juice yield [26]. In addition, it was foreseen that the pectin content had a stronger impact on the yield than the moisture content. For instance, although Muscaris and Regent did not have the lowest moisture content compared to other varieties, they produced the least juice, ranging from 40% to 60%, due to their elevated pectin content (>0.9%). Conversely, Aurora and Golubok, characterized by having the lowest pectin content (<0.8%), yielded the highest juice production, reaching up to 70% and 75%, respectively.

### 3.2. Physicochemical Properties

The basic physicochemical properties of the 20 grape juices are expressed in Table 3. Physicochemical analysis offers valuable insights into the distinct characteristics of grape juice derived from various grape varieties and treatments. Based on a statistical analysis (*p* < 0.05), notable distinctions were observed between the samples. In general, Aurora exhibited the highest dry mass content, exceeding 21.5% in all treatments. Conversely, Golubok showed the lowest dry mass content, ranging from 18.9% to 19.5%. Similarly, Bendaali et al. [26] reported that the dry mass content of grape juice was around 20%.

Furthermore, the dry mass content values were correlated with the initial moisture content. A higher initial moisture content corresponded to a lower dry mass content in the juice. For instance, Aurora, with the lowest moisture content, showed the highest dry mass, whereas Golubok, with the highest moisture content, exhibited the lowest dry mass.

Moreover, the total soluble solids of the grape juice samples ranged from 17.45 °Brix to 21.14 °Brix. The sample with the lowest total soluble solids was MDE, whereas the highest was AIY. As expected, there was a robust correlation between total soluble solids and dry mass, with an R square value of 0.9551. This observation suggested that the processing methods did not significantly affect the total soluble solid content or dry mass. On the contrary, both of these characteristics were influenced by the initial moisture content of the raw materials.

Additionally, turbidity and viscosity are pivotal characteristics that reflect the quality of grape juice [27]. Among the grape juice samples, GIE exhibited the highest turbidity at 992 NTU, while RD exhibited the lowest at 152 NTU. Additionally, the viscosity of the grape juice samples varied from 1.93 mPa·s (GD sample) to 6.9 mPa·s (GIE sample). Interestingly, in the present study, neither the destemming process nor the addition of enzymes influenced turbidity. However, the viscosity showed a significant positive correlation with the destemming process (R^2^ = 0.4516). This phenomenon was due to the fact that the destemmed grapes were more susceptible to releasing herbaceous odors and bitter compounds during pressing [28].

The color parameters of the juice are represented in Figure 2. These samples can be categorized into two clusters: the upper cluster comprises white grape varieties, while the lower cluster comprises red grape varieties. The white grape juices showed a significantly high lightness (L), ranging from 40.8 to 73.5, while the L-values of red grape juice ranged from 2.0 to 17.8.

Although the correlation test (*p* < 0.05) indicated no significant influence of the destemming process and enzyme addition on color changes, it is noteworthy that the grape juices of the Regent variety without enzyme addition (RD and RI) were closely aligned with the white grape juice cluster. This phenomenon can be attributed to the fact that the pulp color of Regent grapes is white. However, the addition of enzymes triggered color extraction from the skin, subsequently elevating the a-value (redness) of the obtained juices, specifically RDE and RIE, which increased from 5.7 to 25.3 and 9.9 to 32.4, respectively.

However, this scenario differed in the case of samples derived from the Golubok variety, which possessed red pulp and dark skin. Surprisingly, the addition of enzymes decreased the a-value of the juices by 12.3 and 14.2 for destemmed (GD) and intact (GI) samples, respectively. This phenomenon can be attributed to the enzymatic breakdown of the skin, resulting in the extraction of the dark color, which subsequently imparted a darker hue to the juice. Simultaneously, this process reduced the b-value (yellowness), producing a distinctive dark purple color in the juice. In essence, the color of the grape juice was influenced by the addition of enzymes; however, the fundamental determinant remained the color of the raw material itself. As highlighted by Guler [29], the color of the juice was mainly shaped by the grape used as the raw material.

### 3.3. Total Polyphenol and Antioxidant Capacity

The experimental analysis revealed the significant effects of the type of grape variety, the destemming process and enzyme addition on the total polyphenol content and antioxidant capacity of the juices (as indicated in Table 4). Among the samples, the highest total polyphenol content was observed in GI (16.12 g of gallic acid equivalent (GAE)/100 mL juice), while the lowest was found in AD (5.84 g of GAE/100 mL). Additionally, the greatest antioxidant potential, determined by the ABTS assay, was recorded in RIE (20.83 mmol of Trolox equivalent (TxE)/L juice), while the lowest value was observed in AD (4.05 mmol of TxE/L). Similar trends emerged from the FRAP method, with the highest values observed in RIE (5.30 mmol of TxE/L) and RDE (5.42 mmol of TxE/L) and the lowest value in AD (1.29 mmol of TxE/L).

Moreno-Montoro et al. [30] observed a similar result, indicating that red grape varieties generally possess higher polyphenol content and antioxidant capacity than white grape varieties. Furthermore, enzyme addition exhibited the potential to enhance the total polyphenol content and antioxidant activity, while the destemming process appeared to be inversely proportional to the bioactive properties of the juices. As previously mentioned by Anastasiadi et al. [31], bioactive compounds were present in the stem. Consequently, the destemming process could eliminate these bioactive compounds from the stems.

Furthermore, a correlation matrix between the total polyphenol content determined by the Folin–Ciocalteu method and UPLC, along with the antioxidant activities (ABTS and FRAP), is visualized in Figure 3. In particular, significant positive correlations were observed among almost all analyses, except for the relationship between ABTS and TPC-UPLC (0.2223). The highest correlation (0.7569) was observed for TPC-FC and FRAP, which was expected since both methods rely on the same mechanism [32]. Conversely, TPC-UPLC exhibited the lowest correlation among all methods. This divergence could be attributed to TPC-UPLC relying on chromatography, which inherently employs a somewhat distinct mechanism compared to the other analysis methods.

### 3.4. Individual Polyphenol Content

The polyphenolic compounds in all grape juice samples were identified by UPLC/ESI-Q-TOF-MS analysis; their details are outlined in Table 5. These polyphenols have been provisionally identified, featuring information such as retention time, maximum wavelength, m/z values and MS/MS fragments. Based on the UPLC-MS analysis and the literature references, 25 phenolic compounds have been tentatively identified, encompassing 4 phenolic acids, 11 flavonols, 5 flavan-3-ols and 5 anthocyanins.

Phenolic acids were detected at 320 nm, flavonols at 360 nm, flavan-3-ols at 280 nm and anthocyanins at 520 nm. This comprehensive profile provides valuable information on the polyphenolic composition of grape juices, facilitating a better understanding of their potential health-promoting properties and flavor attributes.

Figure 4 shows the polyphenol profile of the juice samples. Generally, the polyphenol content of red grape juices exceeded that of white grape juices, ranging from 136 mg/L to 1437.57 mg/L and from 36.26 mg/L to 116.35 mg/L, respectively. This observation was consistent with the findings of Moreno-Montoro et al. [30], who reported similar levels of polyphenolic compounds in red and white grape juices.

Enzyme addition enhanced polyphenol extraction, a phenomenon supported by the findings of Dal Magro et al. [14]. Their study demonstrated that the addition of pectinase during grape juice processing increased the polyphenol content due to the improved skin extraction efficiency. The samples were further characterized by their darker color with enzyme addition. Additionally, it should be noted that polyphenol content is generally higher in grape skin compared to the pulp [31]. The destemming process reduced the polyphenol content due to stem removal, which may have a high polyphenol content, as suggested by Anastasiadi et al. [31].

Furthermore, in white grape juice, the predominant polyphenols were flavan-3-ols, followed by phenolic acids. In contrast, red grape juices were richer in flavonols and anthocyanins. This aligned with the report by Toaldo et al. [33], who found no anthocyanins in any of the white grape juices.

The individual polyphenol content of the grape juice samples is presented in Table 6 for anthocyanins, Table 7 for phenolic acids and flavan-3-ols and Table 8 for flavonols. As expected, the anthocyanin content was absent in the white grape juices, whereas it constituted one of the most abundant polyphenol groups in the red grape juices.

The anthocyanin content exhibited significant variations between samples, reflecting its role in imparting color to red grape juice. The anthocyanin profile demonstrated notable diversity across the grape juices. Delphinidin-3-O-(6′-acetyl)-glucoside (D3O6aGlu), Malvidin-3-O-(6″-caffeoyl)-glucoside (M3O6cGlu), Malvidin-3-O-glucoside (M3OGlu), Petunidin-3-O-(6″-p-coumaroyl)-glucoside (P3O6cGlu) and Petunidin-3-O-glucoside (P3OGlu) exhibited contents ranging from 18.12 to 183.01 mg/L, 11.21 to 24.04 mg/L, 2.15 to 2.49 mg/L, 0.56 to 13.10 mg/L and 41.48 to 269.24 mg/L, respectively.

As indicated in Table 6, the anthocyanin content was primarily influenced by grape variety. The Golubok variety showed higher amounts of each anthocyanin in all treatments. Although enzyme addition was shown to increase anthocyanin content, its impact was not as pronounced as the inherent characteristics of the raw materials.

Furthermore, enzyme addition facilitated the extraction of specific anthocyanins that were not present in samples without enzyme addition. In particular, Malvidin-3-O-glucoside (M3OGlu) was only detected in samples with enzyme addition for both grape varieties. Moreover, in the case of the Regent variety, the anthocyanins Petunidin-3-O-(6″-p-coumaroyl)-glucoside (P3O6cGlu) and Petunidin-3-O-glucoside (P3OGlu) were exclusively detected by enzyme-assisted extraction. This underscores the role of enzymes in enhancing the extraction of specific anthocyanins, which further contributes to the diversity and complexity of the polyphenol profile.

The analysis of the grape juice samples revealed the presence of four phenolic acids (as shown in Table 7). The phenolic acid content exhibited statistically significant differences among the samples. Caffeic acid (CafA) was identified in all treatments of the Johanniter and Golubok varieties, with concentrations ranging from 5.24 to 8.89 mg/L. Interestingly, the destemming process and enzyme addition did not influence the concentration of CafA.

Caftaric acid (CftA) was detected in almost all samples except Johanniter, with concentrations ranging from 0.55 mg/L (AI) to 47.93 mg/L (GDE). In the context of white grape juice, the treatments did not affect CftA content. However, enzyme addition contributed to the improvement in CftA concentration in red grape juices. This emphasizes the role of enzyme addition in selectively influencing the content of specific phenolic compounds in specific varieties.

Coutaric acid (CtA) was not detected in the Golubok variety. Among the samples, the highest concentration of CtA was found in MD (29.41 mg/L), while the lowest concentration was observed in RI (3.08 mg/L). Generally, CtA content was higher in white grape juice.

Fertaric acid (FrtA) was identified in all treatments of the Johanniter variety. In contrast, it was only detected in the Aurora and Golubok varieties when enzyme addition was used. Notably, although it was released through enzyme assistance in the Golubok variety, it exhibited the highest concentration among all samples containing FrtA. Specifically, GIE displayed the highest FrtA content (44.80 mg/L) within the Golubok variety, exceeding the concentrations in the highest FrtA-containing samples of Aurora (ADE, 12.97 mg/L) and Johanniter (JDE, 10.26 mg/L). These observations underline the various responses of different grape varieties to enzyme addition and their corresponding impact on phenolic acid composition.

Furthermore, five flavan-3-ols were identified in the grape juice samples: epicatechin (Ep), (−)-epicatechin 3-gallate (Ep3Gll), Procyanidin B2 (PB2), Procyanidin tetramer (PTt) and Procyanidin trimer (PTm). The latter three flavan-3-ols were present in all samples, while epicatechin was absent in Muscaris and Aurora. Similarly, epicatechin 3-gallate was exclusively detected in Muscaris and Golubok.

Regardless of the variety, flavan-3-ol content increased due to enzyme addition. In particular, there were generally no significant changes observed due to the destemming process in most cases. For instance, PTm content in samples JD, JDE, JI and JDI was found to be 3.73 mg/L, 11.27 mg/L, 4.24 mg/L and 13.05 mg/L, respectively. This emphasizes the role of enzyme addition in enhancing the content of flavan-3-ols in various grape varieties and treatment conditions.

In the current study, 11 flavonols were identified in all samples. However, only dHK3Glu, IR3Glu and dHQ35rha were consistently present in all juice samples, with concentrations ranging from 0.14 mg/L (sample RI) to 2.78 mg/L (sample JDE), 0.33 mg/L (sample RI) to 73.17 mg/L (sample GIE) and 0.29 mg/L (sample RI) to 194.75 mg/L (sample GIE), respectively.

In contrast, Q3Glu and Q3Glcr were exclusively detected in Muscaris and Aurora, with mean values of 3.73 mg/L and 1.17 mg/L, respectively. Additionally, K3OGal was observed exclusively in the red grape juice samples, varying from 13.78 mg/L (sample RD) to 142.75 mg/L (sample GDE). Meanwhile, K3OGlu appeared to be derived from the stem, with only samples MI, MIE, AI, AIE, GI and GIE containing it. Similarly, only samples without the destemming process contained Q3rha (MI, MIE, GI, GIE, RI and RIE), with concentrations ranging from 0.43 mg/L to 61.56 mg/L. Similarly, Q3Gal was found exclusively in samples with enzyme addition (MDE, MIE, GDE, GIE, RDE and RIE). This comprehensive analysis underscores the intricate variations in flavonol composition based on grape variety, treatment, and enzyme addition.

As these findings elucidate, the polyphenol profile within grape juices is notably influenced by grape variety and the processing methods used. This indicates that processing can enhance the extraction of polyphenols from existing compounds but does not generate entirely new polyphenols. The intricate relationship between variety, processing, and the resulting polyphenol content is a crucial aspect to consider in understanding the composition and potential health benefits of grape juice.

## 4. Conclusions

The addition of enzymes and the destemming process potentially improved some quality attributes of grape juice. However, at the same time, they decreased some other attributes. Based on the results obtained from the present study, conclusions can be drawn as follows:(1)Regardless of the variety, enzyme addition and the destemming process improved the yield by 20–30% and 10–15%, respectively.(2)The addition of enzymes increased polyphenol content but, at the same time, altered the color of the juice. Meanwhile, the destemming process eliminated the potential health benefits present in the stems.(3)Although similar treatments were applied, the total polyphenol content and antioxidant capacity of red grape juice were higher than those of white grape juice. This indicates that the characteristics of each variety play an important role in determining the health properties of the juice.(4)A total of 25 polyphenolic compounds were identified in the grape juices. However, the polyphenolic profile was dependent on variety and processing.(5)In summary, this study contributes significantly to understanding how specific grape varieties and processing techniques collectively influence the quality of grape juice. These insights can potentially guide and advance the production of grape juice, leading to improved quality and health-enhancing benefits, especially relevant in challenging climatic conditions similar to those encountered in Poland.

## Figures and Tables

**Figure 1 foods-12-03475-f001:**
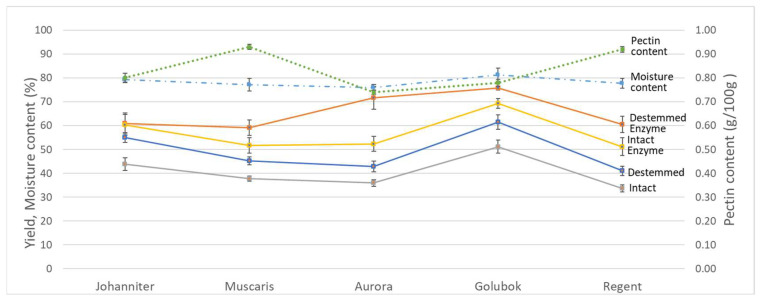
Yield, moisture and pectin content of the grape juice samples. The error bars represent the standard deviation.

**Figure 2 foods-12-03475-f002:**
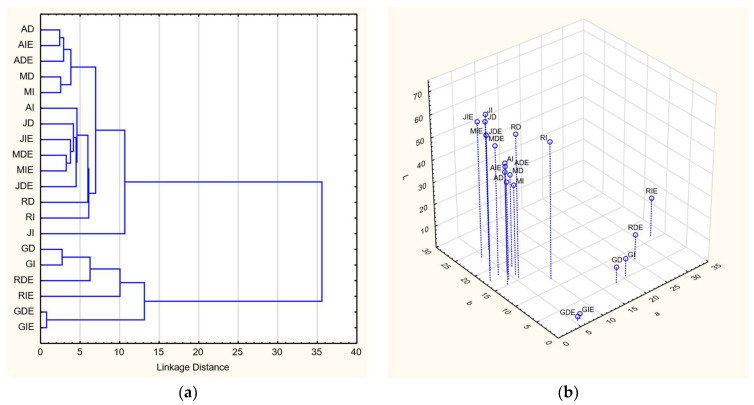
Cluster analysis (**a**) and 3D plot color parameters (**b**) of grape juice samples.

**Figure 3 foods-12-03475-f003:**
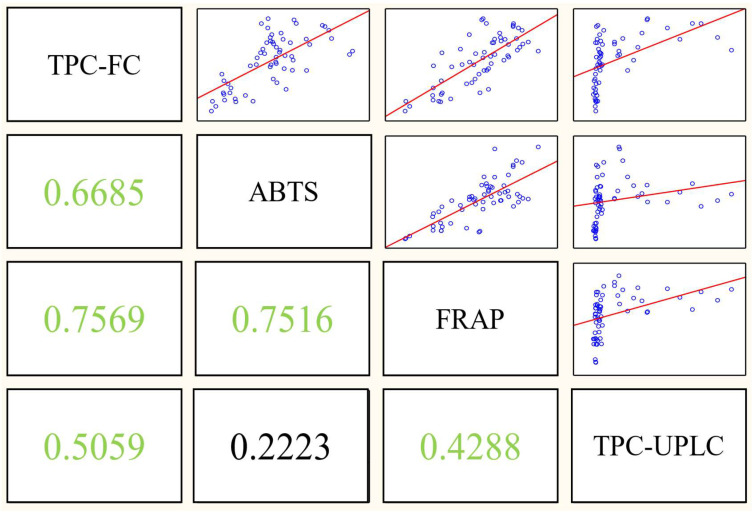
The correlation matrices of the four assays (TPC-FC, ABTS, FRAP and TPC-UPLC), colored green, indicate a significant correlation at *p* < 0.05.

**Figure 4 foods-12-03475-f004:**
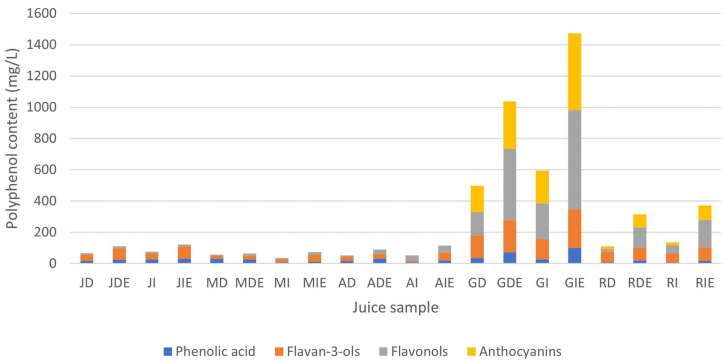
Total polyphenol content of grape juice samples according to the phenolic group.

**Table 1 foods-12-03475-t001:** Characteristics of the hybrid cultivars used in this study.

Cultivars	Parents	Skin Color	Pulp Color
Johanniter	Riesling and Seyve-Villard × Ruländer × Gutedel	Yellowish green	White
Muscaris	Muskateller × Solaris	Green	White
Aurora	Seibel 788 × Seibel 29	Green	White
Golubok	Severnyy × pollen from different varieties: 40 Let Okyabrya, Odesskiy Ranniy and No. 1-17-54 (Alicante Bouschet and Cabernet Sauvignon)	Dark	Dark
Regent	Diana × Chambourcin	Dark	White

**Table 2 foods-12-03475-t002:** Grape juice samples.

Sample	JD	JDE	JI	JIE	MD	MDE	MI	MIE	AD	ADE	AI	AIE	GD	GDE	GI	GIE	RD	RDE	RI	RIE
Variety	Johanniter				Muscaris				Aurora				GoIubok				Regent			
Grape Type	White				White				White				Red				Red			
**Destemmed**	Yes	Yes	No	No	Yes	Yes	No	No	Yes	Yes	No	No	Yes	Yes	No	No	Yes	Yes	No	No
**Enzyme**	No	Yes	No	Yes	No	Yes	No	Yes	No	Yes	No	Yes	No	Yes	No	Yes	No	Yes	No	Yes

**Table 3 foods-12-03475-t003:** Physicochemical properties of grape juices ^‡^.

Sample	Dry Mass	TA	TSS	Turbidity	Viscosity
JD	19.63 ± 0.01 ^efg^	0.73 ± 0.05 ^a^	18.92 ± 0.32 ^c^	297 ± 31 ^abcd^	5.95 ± 0.44 ^gh^
JDE	19.22 ± 0.15 ^cdef^	0.87 ± 0.07 ^a^	18.60 ± 0.25 ^bc^	366 ± 49 ^cd^	3.90 ± 0.31 ^cde^
JI	19.71 ± 0.00 ^fg^	0.65 ± 0.05 ^a^	19.05 ± 0.30 ^c^	197 ± 14 ^ab^	5.24 ± 0.36 ^efg^
JIE	21.32 ± 0.06 ^h^	0.79 ± 0.07 ^a^	20.21 ± 0.33 ^d^	214 ± 12 ^abc^	3.11 ± 0.28 ^abcd^
MD	19.10 ± 0.02 ^cde^	0.58 ± 0.04 ^a^	18.53 ± 0.22 ^bc^	179 ± 11 ^ab^	5.97 ± 0.54 ^gh^
MDE	17.86 ± 0.08 ^a^	0.75 ± 0.07 ^a^	17.45 ± 0.21 ^a^	233 ± 28 ^abcd^	4.21 ± 0.21 ^de^
MI	19.71 ± 0.24 ^fg^	0.53 ± 0.04 ^a^	19.07 ± 0.28 ^c^	190 ± 15 ^ab^	5.65 ± 0.39 ^gh^
MIE	19.37 ± 0.05 ^cdef^	0.69 ± 0.05 ^a^	18.74 ± 0.30 ^c^	231 ± 16 ^abcd^	3.04 ± 0.24 ^abcd^
AD	21.58 ± 0.14 ^hi^	0.63 ± 0.05 ^a^	20.82 ± 0.33 ^de^	855 ± 63 ^f^	6.30 ± 1.10 ^gh^
ADE	21.18 ± 0.15 ^h^	0.63 ± 0.05 ^a^	20.35 ± 0.30 ^de^	658 ± 73 ^e^	3.34 ± 0.15 ^bcd^
AI	21.47 ± 0.13 ^h^	0.49 ± 0.04 ^a^	20.78 ± 0.28 ^de^	317 ± 28 ^bcd^	1.96 ± 0.22 ^ab^
AIE	21.95 ± 0.01 ^i^	0.63 ± 0.05 ^a^	21.14 ± 0.30 ^e^	393 ± 30 ^g^	3.96 ± 0.26 ^cde^
GD	19.03 ± 0.04 ^cd^	0.73 ± 0.07 ^a^	18.72 ± 0.24 ^bc^	905 ± 85 ^f^	1.93 ± 0.30 ^a^
GDE	18.92 ± 0.05 ^bc^	0.69 ± 0.06 ^a^	18.74 ± 0.30 ^c^	915 ± 73 ^f^	6.90 ± 0.47 ^i^
GI	18.92 ± 0.00 ^bc^	0.71 ± 0.05 ^a^	18.48 ± 0.26 ^bc^	874 ± 68 ^f^	2.97 ± 0.44 ^abcd^
GIE	19.46 ± 0.02 ^cdef^	0.65 ± 0.05 ^a^	19.09 ± 0.27 ^c^	992 ± 11 ^f^	4.32 ± 0.10 ^def^
RD	18.47 ± 0.10 ^b^	0.69 ± 0.06 ^a^	17.83 ± 0.22 ^ab^	152 ± 16 ^a^	5.83 ± 0.92 ^gh^
RDE	19.50 ± 0.23 ^def^	0.67 ± 0.05 ^a^	19.00 ± 0.26 ^c^	896 ± 89 ^f^	5.10 ± 0.41 ^efg^
RI	20.08 ± 0.37 ^g^	0.71 ± 0.06 ^a^	19.15 ± 0.30 ^c^	221 ± 33 ^abc^	2.75 ± 0.22 ^abc^
RIE	19.24 ± 0.07 ^cdef^	0.71 ± 0.07 ^a^	18.68 ± 0.27 ^bc^	865 ± 11 ^f^	2.55 ± 0.35 ^abc^

^‡^ value ± SD are means of three repetitions; mean values followed by different letters are statistically different at *p* < 0.05 according to the Tukey post hoc test. Dry mass (%); TA (titratable acidity, g of tartaric acid/100 mL); TSS (total soluble solid, Brix); turbidity (NTU); viscosity (mPa·s).

**Table 4 foods-12-03475-t004:** Physicochemical properties of the grape juices ^‡^.

Sample	TPC (g GAE/100 mL)	TEAC ABTS (mmol TxE/L)	FRAP (mmol TxE/L)
JD	13.17 ± 1.04 ^efg^	12.30 ± 1.52 ^ef^	4.63 ± 0.33 ^defg^
JDE	11.38 ± 0.98 ^cdef^	10.08 ± 1.26 ^cde^	2.59 ± 0.33 ^bc^
JI	15.32 ± 0.95 ^g^	17.42 ± 1.30 ^hi^	4.95 ± 0.39 ^efg^
JIE	11.53 ± 0.75 ^cdef^	10.76 ± 1.28 ^def^	3.83 ± 0.53 ^cdef^
MD	6.95 ± 0.48 ^ab^	6.18 ± 0.68 ^abc^	2.57 ± 0.32 ^bc^
MDE	8.67 ± 1.17 ^abcd^	5.47 ± 0.40 ^ab^	3.79 ± 0.27 ^cde^
MI	8.15 ± 0.85 ^abc^	5.72 ± 0.51 ^ab^	2.45 ± 0.17 ^ab^
MIE	6.68 ± 0.44 ^ab^	8.08 ± 1.10 ^bcd^	2.40 ± 0.15 ^ab^
AD	5.84 ± 0.61 ^a^	4.05 ± 0.30 ^a^	1.29 ± 0.09 ^a^
ADE	12.93 ± 1.88 ^efg^	11.12 ± 0.63 ^def^	3.52 ± 0.25 ^cde^
AI	9.24 ± 1.29 ^abcd^	12.73 ± 0.89 ^efg^	4.11 ± 0.28 ^defg^
AIE	12.81 ± 1.07 ^efg^	11.18 ± 1.43 ^def^	4.31 ± 0.37 ^defg^
GD	11.70 ± 1.45 ^def^	14.12 ± 1.93 ^fgh^	4.79 ± 0.65 ^defg^
GDE	15.33 ± 1.02 ^g^	12.89 ± 1.82 ^efg^	4.82 ± 0.59 ^efg^
GI	16.12 ± 1.00 ^g^	12.72 ± 1.65 ^eg^	4.26 ± 0.37 ^defg^
GIE	14.47 ± 1.66 ^fg^	11.38 ± 1.19 ^def^	5.08 ± 0.32 ^fgh^
RD	10.15 ± 1.30 ^bcde^	12.80 ± 1.19 ^efg^	3.82 ± 0.34 ^cdef^
RDE	13.21 ± 0.92 ^efg^	12.21 ± 0.88 ^ef^	5.42 ± 0.39 ^h^
RI	13.64 ± 1.63 ^efg^	16.43 ± 2.37 ^gh^	4.95 ± 0.58 ^efg^
RIE	13.11 ± 1.05 ^efg^	20.84 ± 1.45 ^i^	5.30 ± 0.80 ^g^

^‡^ value ± SD are means of three repetitions; mean values followed by different letters are statistically different at *p* < 0.05 according to the Tukey post hoc test.

**Table 5 foods-12-03475-t005:** Identification of polyphenols in grape juice samples by UPLC/ESI-Q-TOF-MS.

Tentative Identification	Rt (min)	UV Max (nm)	[M-H]- (*m*/*z*)	MS/MS (*m*/*z*)	Group
Caffeic acid	0.94	327/241	179.0506	-	Phenolic acid
Caftaric acid	3.26	327/241	311.0339	179.0277	Phenolic acid
Coutaric acid	4.15	326/218	295.0368	163.0318	Phenolic acid
Fertaric acid	4.62	326/241	325.0469	193.0431/149.0014	Phenolic acid
Kaempferol-3-O-rutinoside	2.99	343/240	593.1319	285.0628	Flavonols
Quercetin-3-rutinoside	6.66	350/251	609.1573	301.0521	Flavonols
Kaempferol 3-O-galactoside	6.79	351/255	447.2055	285.1445	Flavonols
Quercetin-3-glucuronide	6.90	350/241	477.065	301.0311	Flavonols
Quercetin-3-galactoside	7.04	352/255	463.0869	301.0279	Flavonols
Quercetin-3-glucoside	7.35	348/252	463.2199	301.0455	Flavonols
Quercetin-3-rhamnoside	7.64	342/241	447.0904	301.0325	Flavonols
Kaempferol-3-O-glucoside	7.71	347/241	447.0914	285.0387	Flavonols
Isorhamnetin 3-glucoside	8.12	352	477.1172	315.1055	Flavonols
Dihydroquercetin-3,5-rhamnoside	8.32	346/242	449.1120	303.0239	Flavonols
Dihydrokaempferol-3-glucoside	8.75	346/242	285.0774	257.2076	Flavonols
Procyanidin trimer	2.90	278	865.2130	575.1028/289.1090	Flavan-3-ols
Procyanidin B2	4.00	279	577.1296	289.0646	Flavan-3-ols
Procyanidin tetramer	4.56	280	1153.2663	865.2170/577.1296/289.0458	Flavan-3-ols
(−)-epicatechin	5.21	280	289.0646	-	Flavan-3-ols
(−)-epicatechin 3-gallate	5.93	280	441.1541	289.0646	Flavan-3-ols
Petunidyn-3-O-glucoside	4.00	525/245	479.0999	317.0625	Anthocyanins
Malvidyn-3-O-glucoside	4.48	521/277	493.1141	331.0829	Anthocyanins
Delphinidyn-3-O-(6′-acetyl)-glucoside	5.17	525/277	507.1306	303.0517	Anthocyanins
Petunidyn-3-O-(6″-p-coumaroyl)-glucoside	7.83	529/280	625.1394	317.0465	Anthocyanins
Malvidyn-3-O-(6″-caffeoyl)-glucoside	8.40	531/283	655.1709	331.0641	Anthocyanins

**Table 6 foods-12-03475-t006:** Anthocyanin content of the grape juice samples (mg/L of juice) ^‡^.

Sample	Anthocyanins				
	D3O6aGlu	M3O6cGlu	M3OGlu	P3O6cGlu	P3OGlu
GD	50.52 ± 6.26 ^a^	nd	nd	6.36 ± 0.81 ^b^	112.00 ± 0.77 ^b^
GDE	94.75 ± 7.55 ^d^	nd	2.49 ± 0.20 ^a^	13.24 ± 2.19 ^c^	191.20 ± 10.69 ^c^
GI	69.46 ± 5.39 ^c^	nd	nd	8.66 ± 0.54 ^b^	131.22 ± 11.54 ^b^
GIE	183.01 ± 11.78 ^e^	24.04 ± 1.70 ^b^	2.47 ± 0.37 ^a^	13.10 ± 0.96 ^c^	269.24 ± 16.84 ^d^
RD	18.23 ± 2.87 ^b^	nd	nd	nd	nd
RDE	37.64 ± 4.18 ^a^	nd	2.15 ± 0.30 ^a^	1.19 ± 0.07 ^a^	41.48 ± 4.92 ^a^
RI	18.12 ± 3.11 ^b^	nd	nd	nd	nd
RIE	36.99 ± 5.42 ^a^	11.21 ± 0.24 ^a^	2.30 ± 0.33 ^a^	0.56 ± 0.05 ^a^	43.40 ± 6.06 ^a^

^‡^ value ± SD are means of three repetitions; mean values followed by different letters are statistically different at *p* < 0.05 according to the Tukey post hoc test; Abbreviations: D3O6aGlu—Delphinidyn-3-O-(6′-acetyl)-glucoside; M3O6cGlu—Malvidyn-3-O-(6″-caffeoyl)-glucoside; M3OGlu—Malvidyn-3-O-glucoside; P3O6cGlu—Petunidyn-3-O-(6″-p-coumaroyl)-glucoside; P3OGlu—Petunidyn-3-O-glucoside; nd—not detected.

**Table 7 foods-12-03475-t007:** Phenolic acid and flavan-3-ol content of the grape juice samples (mg/L of juice) ^‡^.

Sample	Phenolic Acids				Flavan-3-ols				
	CafA	CftA	CotA	FrtA	Ep	Ep3Gll	PB2	PTt	PTm
JD	5.24 ± 0.05 ^b^	nd	7.93 ± 0.76 ^b^	3.71 ± 0.29 ^c^	14.87 ± 2.17 ^ab^	nd	4.43 ± 0.64 ^ab^	19.05 ± 1.38 ^c^	3.73 ± 0.08 ^ab^
JDE	6.80 ± 0.26 ^cd^	nd	8.02 ± 1.36 ^b^	10.32 ± 0.93 ^a^	10.05 ± 0.38 ^a^	nd	10.19 ± 1.48 ^bc^	42.97 ± 5.97 ^d^	11.27 ± 0.21 ^bcd^
JI	6.00 ± 0.82 ^bc^	nd	16.83 ± 0.13 ^c^	3.78 ± 0.28 ^c^	21.10 ± 1.80 ^bc^	nd	3.85 ± 0.34 ^ab^	11.05 ± 1.40 ^abc^	4.24 ± 0.57 ^abc^
JIE	5.83 ± 0.35 ^bc^	nd	15.29 ± 1.21 ^c^	10.26 ± 1.60 ^a^	11.71 ± 0.86 ^a^	nd	9.26 ± 0.86 ^abc^	44.44 ± 7.16 ^d^	13.05 ± 1.16 ^cde^
MD	nd	1.12 ± 0.09 ^a^	29.41 ± 1.57 ^d^	nd	nd	0.80 ± 0.07 ^a^	12.83 ± 0.19 ^cde^	3.95 ± 0.16 ^a^	3.17 ± 0.05 ^ab^
MDE	nd	1.95 ± 0.1 ^a^	25.68 ± 3.82 ^d^	nd	nd	1.27 ± 0.1 ^a^	12.93 ± 0.90 ^dce^	4.54 ± 0.29 ^a^	5.22 ± 0.42 ^ab^
MI	nd	0.92 ± 0.01 ^a^	7.81 ± 0.57 ^b^	nd	nd	0.53 ± 0.06 ^a^	14.41 ± 1.31 ^cde^	3.25 ± 0.21 ^a^	3.26 ± 0.34 ^ab^
MIE	nd	1.35 ± 0.12 ^a^	12.23 ± 1.34 ^bc^	nd	nd	0.97 ± 0.01 ^a^	29.54 ± 2.57 ^i^	4.79 ± 0.50 ^a^	8.41 ± 0.54 ^abc^
AD	nd	0.94 ± 0.02 ^a^	17.71 ± 1.82 ^c^	nd	nd	nd	20.37 ± 1.51 ^fgh^	2.84 ± 0.21 ^a^	3.96 ± 0.37 ^ab^
ADE	nd	0.81 ± 0.13 ^a^	16.6 ± 2.55 ^c^	12.97 ± 2.25 ^ab^	nd	1.08 ± 0.07 ^a^	17.74 ± 1.1 ^efg^	7.14 ± 0.59 ^ab^	5.09 ± 0.70 ^abc^
AI	nd	0.55 ± 0.03 ^a^	7.57 ± 0.12 ^b^	nd	nd	nd	3.02 ± 0.23 ^a^	4.84 ± 0.34 ^a^	1.92 ± 0.10 ^a^
AIE	nd	0.89 ± 0.06 ^a^	6.42 ± 0.50 ^ab^	12.83 ± 1.07 ^ab^	28.71 ± 2.82 ^cd^	nd	8.62 ± 1.07 ^abc^	7.78 ± 1.17 ^abc^	5.46 ± 0.4 ^abc^
GD	8.27 ± 0.55 ^a^	28.6 ± 2.77 ^d^	nd	nd	31.11 ± 3.38 ^d^	39.36 ± 3.47 ^b^	25.58 ± 4.16 ^hi^	12.62 ± 1.09 ^abc^	35.44 ± 3.23 ^f^
GDE	8.89 ± 0.56 ^a^	47.93 ± 2.8 ^e^	nd	16.64 ± 1.95 ^b^	13.57 ± 2.00 ^ab^	40.29 ± 6.80 ^b^	15.11 ± 0.32 ^def^	70.13 ± 6.45 ^e^	66.48 ± 2.89 ^g^
GI	7.72 ± 0.61 ^ad^	18.05 ± 1.4 ^c^	nd	nd	28.90 ± 5.21 ^cd^	31.51 ± 2.10 ^b^	26.03 ± 3.92 ^hi^	13.06 ± 1.72 ^abc^	31.28 ± 2.43 ^f^
GIE	8.81 ± 0.50 ^a^	46.69 ± 3.17 ^e^	nd	44.8 ± 3.44 ^d^	14.11 ± 2.24 ^ab^	39.25 ± 6.73 ^b^	12.93 ± 1.98 ^abc^	82.78 ± 11.32 ^b^	99.64 ± 11.74 ^h^
RD	nd	2.91 ± 0.21 ^a^	5.09 ± 0.06 ^a^	nd	32.08 ± 2.36 ^d^	nd	24.87 ± 2.11 ^hi^	7.16 ± 0.57 ^ab^	4.14 ± 0.31 ^ab^
RDE	nd	15.14 ± 1.4 ^bc^	4.87 ± 0.59 ^a^	nd	17.81 ± 3.41 ^ab^	nd	25.87 ± 4.37 ^hi^	18.24 ± 1.68 ^bc^	18.59 ± 2.23 ^e^
RI	nd	2.19 ± 0.17 ^a^	3.08 ± 0.15 ^a^	nd	36.02 ± 4.45 ^d^	nd	23.19 ± 1.75ɡ^hi^	1.70 ± 0.15 ^a^	0.81 ± 0.01 ^a^
RIE	nd	11.96 ± 0.87 ^b^	4.91 ± 0.65 ^a^	nd	33.56 ± 5.05 ^d^	nd	27.29 ± 2.76 ^i^	9.16 ± 0.67 ^ab^	16.21 ± 0.18 ^de^

^‡^ value ± SD are means of three repetitions; mean values followed by different letters are statistically different at *p* < 0.05 according to the Tukey post hoc test; Abbreviations: CafA—Caffeic acid; CftA—Caftaric acid; CotA—Coutaric acid; FrtA—Fertaric acid; Ep—(−)-epicatechin; Ep3Gll—(−)-epicatechin 3-gallate; PB2—Procyanidin B2; PTt—Procyanidin tetramer; PTm—Procyanidin trimer; nd—not detected.

**Table 8 foods-12-03475-t008:** Flavonol content of the grape juice samples (mg/L of juice) ^‡^.

Sample	Flavonols										
	dHK3Glu	IR3Glu	K3OGal	K3OGlu	K3Orut	Q3Gal	Q3Glu	Q3rut	Q3Glcr	Q3rha	dHQ35rha
JD	2.13 ± 0.15 ^fg^	2.92 ± 0.27 ^a^	nd	nd	nd	nd	nd	nd	nd	nd	1.84 ± 0.27 ^a^
JDE	2.78 ± 0.25 ^h^	2.13 ± 0.14 ^a^	nd	nd	nd	nd	nd	nd	nd	nd	6.36 ± 0.40 ^a^
JI	0.45 ± 0.04 ^ab^	3.4 ± 0.50 ^a^	nd	nd	nd	nd	nd	nd	nd	nd	3.91 ± 0.31 ^a^
JIE	2.74 ± 0.39 ^h^	3.34 ± 0.47 ^a^	nd	nd	nd	nd	nd	nd	nd	nd	5.52 ± 0.44 ^a^
MD	0.22 ± 0.03 ^ab^	0.65 ± 0.11 ^a^	nd	nd	0.51 ± 0.05 ^a^	nd	3.08 ± 0.28 ^a^	nd	nd	nd	0.74 ± 0.10 ^a^
MDE	0.98 ± 0.06 ^cd^	0.69 ± 0.09 ^a^	nd	nd	2.24 ± 0.02 ^a^	1.11 ± 0.20 ^b^	4.30 ± 0.23 ^b^	nd	nd	nd	2.21 ± 0.23 ^a^
MI	0.32 ± 0.02 ^ab^	0.61 ± 0.05 ^a^	nd	0.05 ± 0.00 ^a^	0.53 ± 0.08 ^a^	nd	3.20 ± 0.45 ^a^	nd	nd	0.43 ± 0.03 ^a^	0.95 ± 0.07 ^a^
MIE	1.69 ± 0.13 ^ef^	0.7 ± 0.12 ^a^	nd	0.95 ± 0.05 ^a^	2.38 ± 0.29 ^a^	1.17 ± 0.02 ^b^	4.34 ± 0.54 ^b^	nd	nd	0.71 ± 0.07 ^a^	3.15 ± 0.39 ^a^
AD	0.38 ± 0.03 ^ab^	0.87 ± 0.09 ^a^	nd	nd	3.88 ± 0.58 ^a^	nd	nd	nd	1.24 ± 0.15 ^a^	nd	0.38 ± 0.02 ^a^
ADE	0.71 ± 0.04 ^bc^	2.01 ± 0.06 ^a^	nd	nd	24.28 ± 3.29 ^d^	nd	nd	nd	1.12 ± 0.14 ^a^	nd	1.22 ± 0.10 ^a^
AI	0.41 ± 0.07 ^ab^	0.71 ± 0.04 ^a^	nd	0.46 ± 0.06 ^a^	9.88 ± 1.62 ^b^	nd	nd	19.56 ± 2.34 ^a^	1.22 ± 0.03 ^a^	nd	1.26 ± 0.17 ^a^
AIE	0.69 ± 0.10 ^bc^	1.64 ± 0.16 ^a^	nd	1.03 ± 0.14 ^a^	36.85 ± 0.56 ^e^	nd	nd	nd	1.11 ± 0.07 ^a^	nd	4.33 ± 0.26 ^a^
GD	2.05 ± 0.05 ^fg^	12.92 ± 1.11 ^b^	103.62 ± 13.87 ^b^	nd	17.98 ± 1.02 ^c^	nd	nd	nd	nd	nd	1.83 ± 0.13 ^a^
GDE	2.30 ± 0.29 ^fh^	69.33 ± 6.77 ^d^	142.75 ± 4.01 ^c^	nd	26.45 ± 1.87 ^d^	70.52 ± 9.45 ^a^	nd	nd	nd	nd	143.40 ± 17.09 ^b^
GI	1.37 ± 0.08 ^de^	1.22 ± 0.08 ^a^	107.75 ± 1.11 ^b^	43.2 ± 3.52 ^b^	24.22 ± 3.20 ^d^	nd	nd	16.94 ± 1.29 ^a^	nd	31.22 ± 1.01 ^b^	3.53 ± 0.32 ^a^
GIE	1.93 ± 0.25 ^fg^	73.17 ± 5.73 ^d^	131.74 ± 8.93 ^c^	66.42 ± 5.20 ^c^	34.23 ± 2.68 ^e^	68.91 ± 3.90 ^a^	nd	nd	nd	61.56 ± 4.34 ^d^	194.75 ± 14.23 ^c^
RD	0.43 ± 0.06 ^ab^	0.33 ± 0.02 ^a^	13.78 ± 1.93 ^a^	nd	0.45 ± 0.06 ^a^	nd	nd	nd	nd	nd	0.30 ± 0.02 ^a^
RDE	1.39 ± 0.10 ^de^	26.49 ± 1.50 ^c^	16.74 ± 1.93 ^a^	nd	10.57 ± 0.80 ^b^	72.63 ± 9.24 ^a^	nd	nd	nd	nd	3.95 ± 0.26 ^a^
RI	0.14 ± 0.01 ^a^	0.33 ± 0.04 ^a^	13.80 ± 1.62 ^a^	nd	0.45 ± 0.02 ^a^	nd	nd	29.36 ± 2.95 ^b^	nd	6.98 ± 0.64 ^c^	0.29 ± 0.02 ^a^
RIE	1.89 ± 0.26 ^fg^	24.10 ± 3.32 ^c^	19.26 ± 2.57 ^a^	nd	18.11 ± 2.80 ^c^	76.20 ± 9.63 ^a^	nd	nd	nd	33.60 ± 2.42 ^b^	1.79 ± 0.13 ^a^

^‡^ value ± SD are means of three repetitions; mean values followed by different letters are statistically different at *p* < 0.05 according to the Tukey post hoc test; Abbreviations: dHK3Glu—Dihydrokaempferol-3-glucoside; IR3Glu—Isorhamnetin 3-glucoside; K3OGal—Kaempferol-3-O-galactoside; K3OGlu—Kaempferol-3-O-glucoside; K3Orut—Kaempferol-3-O-rutinoside; Q3Gal—Quercetin-3-galactoside; Q3Glu—Quercetin-3-glucoside; Q3rut—Quercetin-3-rutinoside; Q3Glcr—Quercetin-3-glucuronide; Q3rha—Quercetin-3-rhamnoside; dHQ35rha—Dihydroquercetin-3,5-rhamnoside; nd—not detected.

## Data Availability

Data is contained within the article.

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
