# Peer review of "Influence of Variety, Enzyme Addition and Destemming on Yield and Bioactive Compounds of Juices from Selected Hybrid Grape Varieties Cultivated in Poland"

_foods, 2023, doi:10.3390/foods12183475_

Round 1
Reviewer 1 Report
The work is good but needs some corrections
Comment on Manuscripts
“ Influence of variety, enzyme addition and destemming on 2 yield and bioactive compound of the juices from selected hy- 3 brid grapes varieties cultivated in Poland”
1. Abstract is good but more findings of the research paper can be included and General statement. “Poland stands out as one of the coldest regions for grape cultivation, with its weather proving unsuitable for conventional grape varieties due to their vulnerability to winter frost and fungal diseases. A hybrid grape is a crossbreed grapevine between two different varieties that can 14 address the issue. Prior studies using hybrid grapes have predominantly delved into wine processing. A notable gap exists in research exploring the processing of hybrid grape juice using locally 16 grown materials.” should be cut short or deleted
2. . Introduction is written nicely but lack the proper justification of present work, where is methodology adopted with reference to other studied or not should be added. Whereas Author have complete the justification in one line as given “Several previous studies on hybrid grapes grown in Poland investigated mostly wine 61 [9,11–13] and no research was found that investigated juice processing using hybrid 62 grapes grown in Poland as raw materials.”
3. In the manuscript material and method section author have written “Before macerated them at 40 oC for 90 minutes the grapes were crushed and 0.05% (w/w) 92 of Pectinex enzyme was added. The must then was pressed using a hydraulic press and 93 filtered to obtain juice. The juices were immediately frozen at -20 °C until analysis” Make model of instrument should be added and why is was immediately frozen at -20 °C? any specific reason.
4. Section 2.4 “The dry matter content (dm) of the obtained juices was carried out in a vacuum dryer (SPT-200, ZEAMiL Horyzont, Kraków, Poland) at 80 °C for 24 h at the pressure of 1 kPa. The measurement was done in duplicate” Here author is calculating dry mass as per my understanding it is moisture content analysis, then how moisture content is analysed in vacuum dryer
5. In the Colour analysis 2.5 for the samples Hue angle and Chroma analysis has not done? add it also Needs reference add below reference.
Utilization of lima bean starch as an edible coating base material for sapota fruit shelf-life enhancement. Journal of Agriculture and Food Research, 12,1-9
6. Figure.1.Author has given several graphs, please explain the relevant of such complex graghs as these graphs are not in continues form and very difficult to understand from a point of view of readers.
7. Stastical analysis given by author is very good and neatly explained.
8. Section 2.7,2.8,2.9 and 2.10 and 3.3,3.4 and 3.5 are well explained and results are clear to understand
9. As per the results findings the Conclusion given is bit short, it should be a complete gist of study in bullet points which is easy to understand for the the viewers.
ok
Author Response
Dear Reviewer,
First of all I would like to thank you for giving us the opportunity to submit a revised draft of our manuscript entitled “Influence of variety, enzyme addition and destemming on yield and bioactive compound of the juices from selected hybrid grapes varieties cultivated in Poland”. We made efforts to prepare point-by-point response to your comments and concerns (all the revisions are marked up using the track changes):
- Abstract is good but more findings of the research paper can be included and General statement. “Poland stands out as one of the coldest regions for grape cultivation, with its weather proving unsuitable for conventional grape varieties due to their vulnerability to winter frost and fungal diseases. A hybrid grape is a crossbreed grapevine between two different varieties that can 14 address the issue. Prior studies using hybrid grapes have predominantly delved into wine processing. A notable gap exists in research exploring the processing of hybrid grape juice using locally 16 grown materials.” should be cut short or deleted
The abstract has been revised
- . Introduction is written nicely but lack the proper justification of present work, where is methodology adopted with reference to other studied or not should be added. Whereas Author have complete the justification in one line as given “Several previous studies on hybrid grapes grown in Poland investigated mostly wine 61 [9,11–13] and no research was found that investigated juice processing using hybrid 62 grapes grown in Poland as raw materials.”
The introduction has been modified
- In the manuscript material and method section author have written “Before macerated them at 40 oC for 90 minutes the grapes were crushed and 0.05% (w/w) 92 of Pectinex enzyme was added. The must then was pressed using a hydraulic press and 93 filtered to obtain juice. The juices were immediately frozen at -20 °C until analysis” Make model of instrument should be added and why is was immediately frozen at -20 °C? any specific reason.
The model of instrument has been added.
The juices were immediately frozen to preserve the juices before further analysis because all the analysis cannot be done in one working day.
- Section 2.4 “The dry matter content (dm) of the obtained juices was carried out in a vacuum dryer (SPT-200, ZEAMiL Horyzont, Kraków, Poland) at 80 °C for 24 h at the pressure of 1 kPa. The measurement was done in duplicate” Here author is calculating dry mass as per my understanding it is moisture content analysis, then how moisture content is analysed in vacuum dryer
The methodology of measuring dry matter (DM) and moisture content (MC) is similar because in wet basis, DM=100-MC
- In the Colour analysis 2.5 for the samples Hue angle and Chroma analysis has not done? add it also Needs reference add below reference.
Utilization of lima bean starch as an edible coating base material for sapota fruit shelf-life enhancement. Journal of Agriculture and Food Research, 12,1-9
The color was measured using CIELab color space where chroma and hue angle are not explicitly represented as separate components. Therefore, the chroma (C*) and hue angle (h*) are not presented in the article although both of them can be calculated based on L-a-b value using the following formula:
C∗=sqrt(a^2+b^2)
and
h*=arctan(b/a)
The reference has been added
- 1.Author has given several graphs, please explain the relevant of such complex graghs as these graphs are not in continues form and very difficult to understand from a point of view of readers.
Figure 1 has been deleted
- Stastical analysis given by author is very good and neatly explained.
Thank you
- Section 2.7,2.8,2.9 and 2.10 and 3.3,3.4 and 3.5 are well explained and results are clear to understand
Thank you
- As per the results findings the Conclusion given is bit short, it should be a complete gist of study in bullet points which is easy to understand for the the viewers.
The conclusion has been reformulated

Reviewer 2 Report
Dear Authors, I have read the MS entitled Influence of variety, enzyme addition and destemming on yield and bioactive compound of the juices from selected hybrid grapes varieties cultivated in Poland.
The article aims at analysing PIWI varieties suitable for colder climate countries and the composition of the grape must influenced by enzyme addition.
I have some observations:
-please check lines 49 and 50 and correct info regarding grapes, musts and wines with higher acidity. Theoretically, these are generally accepted to be a positive aspect, in maintaining the equilibrium of the final products.
-please check that the term "hybrid grapes" is appropriate. Is it not PIWI?
-please correct terminology: "flesh" should be "pulp"
-please correct terminology: grape juice is must
-please use symbols like TRADEMARK or REGISTERED for all commercial dennominations of reagents, enzymes etc
-what is the purpose of figure 1? I don't think it is necessary
-line 92 - grapes were macerated at 40 degrees Celsius? MAybe 4? Please correct if necessary
-is there a control within the analysed samples? Maybe add another major grape variety grown in Poland and compare and contrast with the already selected ones
-line 163- please remove full stop at the beggining of the paragraph
-please check journal requirements regarding how grape varieties should be written. Italics or with inverted comas or followed by cv. Correct, if necessary, all over text.
-please add some conclusions on the found results.
English language needs improvement.
Usually, plural of "grape" should be used. See introduction mainly.
Also, please correct active voice with passive voice. Never use "we" , "our" but "the authors"
-line 31 - please change "grape is" to "grapes are"
-line 35 - please change "grape and its derivatives" to "grapes..."
-line 49- please correct "preserve the acidS"
-line 50 - please correct "acid" grapes
Author Response
Dear Reviewer,
First of all I would like to thank you for giving me the opportunity to submit a revised draft of our manuscript entitled “Influence of variety, enzyme addition and destemming on yield and bioactive compound of the juices from selected hybrid grapes varieties cultivated in Poland”. We made efforts to prepare point-by-point response to your comments and concerns (all the revisions are marked up using the track changes):
- -please check lines 49 and 50 and correct info regarding grapes, musts and wines with higher acidity. Theoretically, these are generally accepted to be a positive aspect, in maintaining the equilibrium of the final products.
The sentence has been revised
- -please check that the term "hybrid grapes" is appropriate. Is it not PIWI?
PIWI grapes are hybrid grape varieties, but not all hybrid grapes are PIWI. In the article, all varieties are hybrids, with three of them classified as PIWI (Johanniter, Muscaris, and Regent), while two varieties, Aurora and Golubok, are not considered PIWI varieties. Therefore, the term “hybrid grape” is appropriate.
- -please correct terminology: "flesh" should be "pulp"
It has been corrected
- -please correct terminology: grape juice is must
Thank you for this feedback. As per our understanding, "Grape juice" refers to the liquid extracted from grapes after they have been crushed and the solids (such as skins, seeds, and stems) have been separated, while “must” refers to the crushed grape mixture that includes not only the juice but also the aforementioned grape solids.
We apologize for any confusion that may have arisen due to our previous description of the process, which led you referred to must. We have revised section 2.3 (Juice processing), to make it clearer that the sample we analyzed were grape juice since the must was filtered.
- -please use symbols like TRADEMARK or REGISTERED for all commercial dennominations of reagents, enzymes etc
The symbol ® has been added
- -what is the purpose of figure 1? I don't think it is necessary
Figure 1 has been deleted
- -line 92 - grapes were macerated at 40 degrees Celsius? MAybe 4? Please correct if necessary
The grape were macerated at 40 (forty) degrees
- -is there a control within the analysed samples? Maybe add another major grape variety grown in Poland and compare and contrast with the already selected ones
The control sample are the sample without enzyme addition and without destemming process. Additionally, Regent and Johanniter are major grape varieties grown in Poland
- -line 163- please remove full stop at the beggining of the paragraph
The full stop has been removed
- -please check journal requirements regarding how grape varieties should be written. Italics or with inverted comas or followed by cv. Correct, if necessary, all over text.
The cultivars were written referred to The Manual of Scientific Style, 2009.
Cultivar names are written in roman type with an initial capital letter and are placed within single quotation marks when they come after the scientific name. Single quotation marks are not needed when the cultivar name is used alone.
- -please add some conclusions on the found results.
The conclusion has been reformulated
- English language needs improvement.
Addressing the issue that extensive English editing is required. The manuscript has been revised by our linguistic expert. All changes are marked up using track changes
- Usually, plural of "grape" should be used. See introduction mainly.
It has been corrected
- Also, please correct active voice with passive voice. Never use "we" , "our" but "the authors"
It has been corrected
- -line 31 - please change "grape is" to "grapes are"
It has been corrected
- -line 35 - please change "grape and its derivatives" to "grapes..."
It has been corrected
- -line 49- please correct "preserve the acidS"
The sentence has been revised
- -line 50 - please correct "acid" grapes
The sentence has been revised
